# Two-Lenses Model to Unfold Sustainability Innovations: A Tool Proposal from Sustainable Business Model and Performance Constructs

Sandra Naomi Morioka [1], Maria Holgado [2], Steve Evans [3], Marly M. Carvalho [4], Paulo Rotella Junior [1] and Ivan Bolis [5,*]

1. Department of Production Engineering, Federal University of Paraiba, João Pessoa 58051-900, Brazil; sandra.morioka@academico.ufpb.br (S.N.M.); paulo.rotella@academico.ufpb.br (P.R.J.)
2. Department of Management, University of Sussex, Brighton BN1 9RH, UK; m.holgado@sussex.ac.uk
3. Institute for Manufacturing, University of Cambridge, Cambridge CB3 0FS, UK; se321@cam.ac.uk
4. Department of Production Engineering, University of São Paulo, São Paulo 05508-900, Brazil; marlymc@usp.br
5. Department of Psychology, Federal University of Paraiba, João Pessoa 58051-900, Brazil
* Correspondence: bolis.ivan@alumni.usp.br

**Abstract:** This research combines corporate sustainability performance and sustainable business model concepts to improve the corporate sustainability of organizations. The main objective of this article is to propose and apply a tool to identify sustainable innovation opportunities through a structured brainstorming process while providing a systemic business perspective and a strong multi-stakeholder orientation. The present qualitative research was carried out in two phases. The first phase consisted of a critical analysis of literature that enabled the proposition of the Two-Lenses Model (2LM) for sustainability innovation. The corporate sustainability performance lens encompasses strategic drivers, business processes, capabilities, stakeholders' satisfaction and contributions. The sustainable business models lens considers value proposition, value creation and delivery system and value capture and sharing. The second phase consists of applying the 2LM in two industrial cases. The results show that the proposed model has the potential to trigger the identification of opportunities through two mechanisms: misalignments between performance dimensions and gaps in stakeholder satisfaction. Further research opportunities lie on deepening into these findings and investigating the implementation process for the identified innovation opportunities.

**Keywords:** sustainable business model; sustainability performance; sustainability innovation; performance prism; corporate sustainability; stakeholder value

## 1. Introduction

Companies aiming not only to maximize profit but also to promote social and environmental value is a compelling idea and hard to put into practice. Multiple approaches on sustainability and sustainable development exist without a common ground [1–4]. In the context of sustainable development, a recent trend is to base decisions on the sustainable development goals (SDGs) proposed by the United Nations. Included in the 2030 Agenda for Sustainable Development, 17 goals and 169 targets point to priorities for peace and prosperity for people and the planet, now and in the future [5]. These objectives focus mainly on creating social and environmental value, being very useful in directing, for example, public policies to create sustainable cities and communities [6]. In companies, the creation of social and environmental value is hampered by the need to act in a market context, characterized by strong economic and financial pressures [7]. Despite this complex environment, there is evidence of interest from companies in acting and implementing corporate sustainability policies try to overcome this challenge.

Corporate sustainability has been defined in many ways. In the present paper, we consider three principles for corporate sustainability to delimit this concept, as previously considered by the literature [8,9], encompassing that sustainability at business level includes: (i) the triple bottom line approach addressing economic, environmental and social goals [10]; (ii) the consideration of internal and external stakeholders in companies' decisions [11,12]; and (iii) the short-, medium- and also long-term impacts for current and future generations [13,14]. We consider that these three principles can support companies to improve their business models and advance towards sustainable development.

Contributions to sustainable development challenges could bring improvements in business performance as argued by the concepts of shared value creation [15], sweet spots [16] and win-win relations between social, environmental and economic aspects [17], as evidence shows that companies' business models can be configured to be able to contribute to global sustainable development and, at the same time, to their respective competitive advantage [8,18]. Various drivers can influence companies in improving their sustainability performance, including leadership and the business case, but also reputation, customer demands and expectations and regulation/legislation [19]. However, sustainability innovations remain unrealized in many cases, and companies find it difficult to innovate towards more sustainable business models (SBMs) [20]. Evidence shows, for instance, that companies, even the leading ones, are still struggling in integrating sustainability into corporate strategy, as they lack deep changes and innovations in their business processes [21]. A good business strategy formulation is important to achieve a distinctive competitive advantage [22].

Innovations towards more SBMs can happen as a constant movement, considering a critical analysis of performance in terms of contributions to sustainable development. This is aligned with the fact that business models need to be in a constant process of transformation and change, given the changing socioenvironmental context and configuration of sustainable development challenges [23]. For example, the impact of moving from the current linear model of economy to a circular one needs many changes [24], needing to consider its barriers and enablers [25]. The design of business model alternatives, their communication and their evaluation are core issues of the innovation process. Sustainable value can be seen as the value created by business that drives both to shareholder value and to a more sustainable world [26]. These authors discussed that the pursuit of sustainable value occurs in the following three steps: diagnosis, opportunity assessment and implementation. The diagnosis step, however, can be challenging for organizations aiming to become more sustainable, since they need to switch the paradigm from seeking solely financial benefits to pursuing also environmental and social goals in the long term. There is a need for a systemic view that considers a global perspective, the different elements and their interconnections [27]. In this line, Corporate Sustainability Performance (CSP) theoretical background could bring a holistic view of the organization and potentially contributing to the diagnosis step.

This study aims to bridge CSP and SBMs foundations to propose and apply a tool to support identifying sustainable innovation opportunities. Providing a more systemic business perspective, this tool seeks to enable a structured brainstorming based on CSP and SBMs concepts to support the identification of sustainability innovation opportunities that include a strong multi-stakeholder perspective, bringing to the discussion stakeholders' needs and contributions to current business models.

For this, we propose to analyse this RQ through two lenses—SBM and CSP—and to engage practitioners in the research as experts in the field. These two lenses were chosen given their complementary contribution to sustainability. On the one hand, SBMs support the integration of sustainability into organizations, since it brings emphasis on sustainability innovations as business opportunities, affecting sustainable value exchange with stakeholders. However, this body of knowledge is relatively recent [28] and still lacks case studies and empirical evidence [29,30]. To contribute to SBM literature, the CSP literature can be an interesting approach, since it is relatively more mature [9,31]

and performance systems have potential to influence corporate results given their direct impact on managers' actions and decisions [32]. This relation was previously explored by Morioka et al. [9], showing that this combination of approaches can bring interesting contributions for theory and practice, enabling a structured organization of the information about company's performance (in the specific dimensions and its relations), incentivizing the consideration of stakeholders beyond customers and shareholders and promoting a critical analysis of the company based on business model elements. A recent study [20] discussed how SBMs are effectively driving sustainable performance, identifying some tensions and conflicts between the three pillars of sustainability when developing and implementing SBMs. This points to the need to further investigate this connection.

In doing so, a Two-Lenses Model (2LM) is proposed, based on a performance measurement system (PMS) framework called the performance prism model [33], and three elements of business models—a value proposition, creation and delivery system and value capture [34]—in the context of corporate sustainability. Presenting two test cases, this paper develops and applies a step-by-step process to support companies in the critical analysis of their current CSP and, based on this, instigate them towards the identification of innovation opportunities to promote SBM. Using an academic perspective to develop and test practice-oriented tools is not novel to the literature on corporate sustainability [35–37]. While some are tools based on computer interface and programming [35,37], others seek to instigate corporate sustainability innovation discussions based on face-to-face interactions between academics and practitioners with support of a brainstorming/workshop tool [36,38,39]. The present research is more related to the second approach on tool proposition. This research proposes to advance knowledge towards two research gaps identified by Evans et al. [30]: the lack of knowledge on drivers for successful sustainable business model innovations and the lack of cases addressing how to innovate business models for sustainability.

## 2. Theoretical Background: Sustainable Business Models and Corporate Sustainability Performance

This section presents the integration of two disciplinary views for sustainability. These lenses are brought by incorporating concepts within the research fields of SBMs and CSP. The SBM lens is derived from the combination of business models and corporate sustainability concepts. The SBM concept is useful to help companies describe, analyze, manage and communicate its value proposition as well as the creation, delivery and capture of value [40], since it characterizes firms' priorities, resources and activities. Therefore, SBM represents how companies exchange sustainable value with their stakeholders [30]. In this sense, SBMs seek not only to promote financial results but also to maximize sustainable value, which can be considered as the economic, environmental and social benefits in the short, medium and long term, specific for each corporate stakeholder. SBMs are concerned with making explicit the mechanisms which logically integrate a company's goals, resources, processes and stakeholders, aiming at promoting sustainable value.

Different configurations of business models can be derived from social, organizational and technological innovations [28,41], generating impact on companies' offerings and/or business processes towards improved corporate sustainability performance [8]. In this sense, business plans for SBM's need to go beyond providing a positive financial business case [11] and need to consider various factors for its success, such as unified organizational vision, a visible on-board leadership, a multi-stakeholder perspective, focus on innovation and communication capabilities to deliver the core message and strategy implementation [42].

The CSP lens covers the concepts of corporate performance and corporate sustainability. Corporate performance has been argued to refer to the efficiency and effectiveness of business actions [43]. This definition implies the presence of goals (or parameters) to delimit what the company is aiming to achieve in an efficient and effective manner. In our context, the approach of CSP is related to corporate sustainability principles, e.g., to firm's contribution to global sustainable development. To ensure that contributions to

corporate sustainability are efficient and effective, organizations need transparency on business decisions and activities, as well as metrics to evaluate and compare their impacts [44]. Therefore, a CSP lens can be used as an instrument to assess the firm's contribution to sustainable development. The literature on CSP has been growing fast in the past decade, addressing the challenges of assessing, improving and reporting CSP [9].

This lens has been also been discussed as environmental, social or sustainability accounting [45] and disclosure [46]. There are various proposals of sustainability indicators and frameworks, such as the Global Reporting Initiative (GRI) indicators based on the triple bottom line approach [47] and the Environmental, Social and Governance (ESG) indicators [48]. Combining sustainability indicators under a specific logic given by their particular organizational context, companies can develop sustainability performance measurement systems (SMPS) [31,49]. SMPS can support companies in planning and controlling their business activities [31]. Reviewing several frameworks for performance measurement systems (PMSs), Yadav et al. [50] pointed out that classical PMSs (such as Balanced Scorecard and Performance Prism) have been evolved over time, including gradually more CSP issues. Examples are the use of Balanced Scorecard [51,52] and the Performance Prism [53] in the corporate sustainability context. Table 1 summarizes the main aspects of the two conceptual lenses combined in this research. As shown in Table 1, these two concepts bring their own theoretical backgrounds within the broader corporate sustainability construct. With specific particularities in application, associated main concepts, orientations and elements, SBM and CSP lenses converge in terms of unit of analysis (the organization) and of objective, which is to help companies make decisions that can contribute to more sustainable, e.g., to intensify organizations' contribution to sustainable development. Thus, the present research argues that the combination of specific aspects of these lenses can provide a more systemic critical perception of the organization and, therefore, supporting the identification of sustainability innovation opportunities. These aspects are explained in the following sections (Sections 2.1 and 2.2), which are then structured into a practical tool presented in Section 2.3. The proposed tool is process-oriented, as it is composed of three sequential stages, Part A, B and C (Figure 1). This step-by-step approach seeks to facilitate the practical implementation of the proposed tool by academics and practitioners.

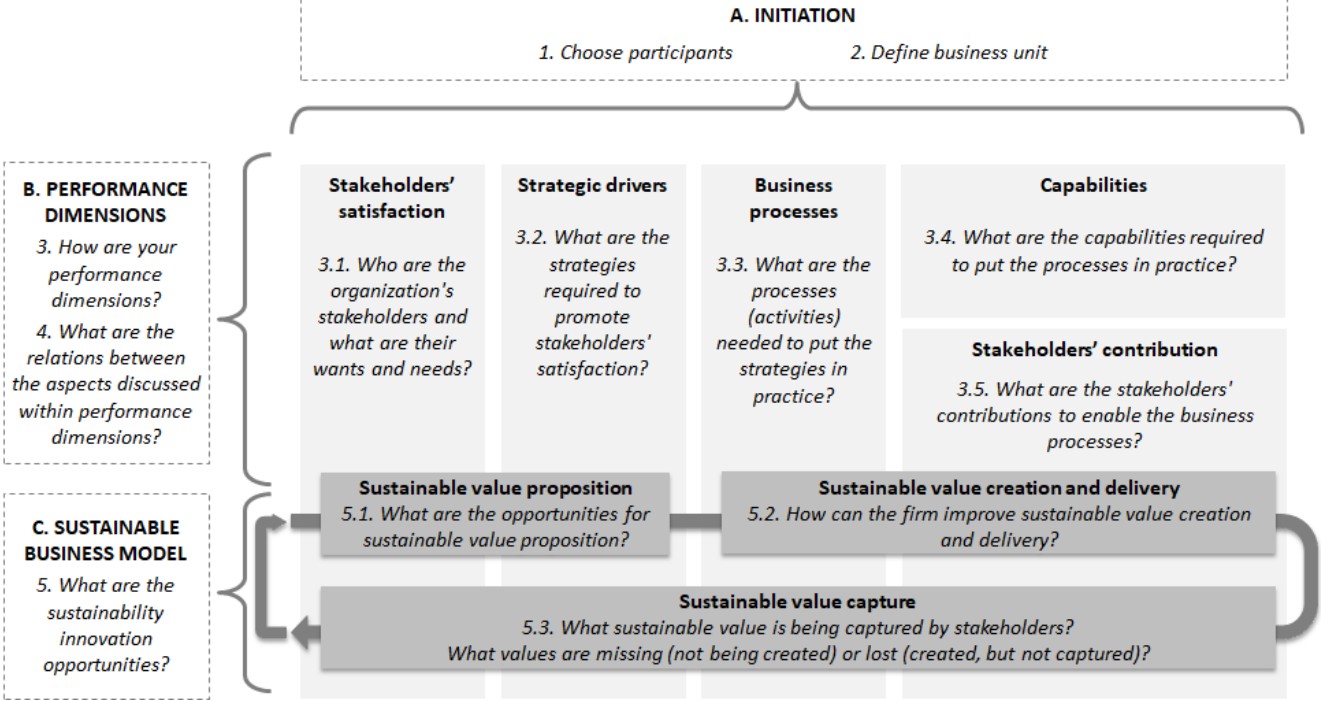

**Figure 1.** Two-Lenses Model (2LM) for sustainability innovation.

**Table 1.** Delimitation of SBM and CSP concepts.

| Aspect | Sustainable Business Model Lens | Corporate Sustainability Performance Lens |
|---|---|---|
| Basic conceptual approaches | Business model (such as Osterwalder et al. [54] and Richardson [34]). + | Performance (such as Bititci et al. [55], Kaplan and Norton [56] and Neely [57]). + |
|  | Corporate sustainability Capacity of organizations to contribute to sustainable development, which includes concern related to the following principles: triple bottom line (economic, environmental and social pillars); multi-stakeholders' interests and timeframes (short, medium and long term) | |
| Unit of analysis | Organization | |
| Aim | To help companies make decisions that can contribute to businesses become more sustainable, e.g., to intensify organizations' contribution to sustainable development | |
| Application | Representing the mechanism of how companies exchange sustainable value with stakeholders. | Representing the situation according the specific parameters, enabling to evaluate the gap between current and aimed situation |
| Main concepts | Sustainable business models Representation of an organization's mechanisms to exchange sustainable value with stakeholders Sustainable value Set of benefits aligned with the principles of corporate sustainability Sustainability innovation Implementation of a new solution capable of improving sustainable value proposition, creation, delivery and capture | Sustainability performance Efficiency or effectiveness of action [43] that contributes to corporate sustainability Sustainability performance indicator Quantification of sustainability performance, according to specific criteria, measurement unit and context Sustainability performance measurement system Set of individual sustainability indicators, organized as a system according to a defined logic and connected to the organizational context (adapted from Neely et al. [43]) |
| Orientation | Stakeholder-centred Depends on the stakeholder (such as goals, needs, context, etc.) | Firm-centred Depends on the company |
| Cross-relation | Sustainability performance indicators are important to quantify sustainable value, in order to assess the gap between current and aimed sustainable value | Knowing what is value for each stakeholders can be used as basis to set sustainability performance criteria and goals |
| Examples of dimensions | - Value propositions; creation and delivery system; value captured [34]<br>- Value proposition; customer interface; infrastructure management; financial aspects [54] | - Prism: stakeholders' satisfaction; strategic drivers; business processes; capabilities; and stakeholders' contributions [33]<br>- Balanced scorecard: financial; customer; internal processes; learning and growth perspectives [58] |

*2.1. Exchanging Sustainable Value with Stakeholders: Sustainable Business Models*

The first lens is represented in the conceptual framework by three elements of business models: value proposition, value creation and delivery system and value capture [34]. Following this, we discuss each business model element in the context of corporate sustainability. The value proposition refers to the company's offerings which are shaped as individual products or services or as bundles of products and services [54]. Den Ouden [59] includes the understanding of the primary users and buyers, their needs and aspirations, the solution offered and the analysis of alternatives and differentiators of the solution as part of the value proposition. From a corporate sustainability perspective, the value proposition includes initiatives that create long-term value for all stakeholders [27]. A sustainable value proposition defines what economic, environmental and social value the company expects to deliver to its stakeholders, providing benefits to satisfy their specific needs or desires.

The value creation and delivery system comprises the firm's resources, capabilities and inter-organizational network [34], which need to be organized and coordinated within a firm's primary and secondary activities [60]. Several initiatives can be implemented by the company to promote the integration of sustainability into operations and production, management and strategy, organizational systems, marketing and procurement, assessment and communication [61]. This includes various approaches, such as cleaner production, design-for-environment, eco-efficiency, environmental and social accounting and ethical investment, among many others [1,61]. The challenge for companies is, therefore, to additionally consider social and environmental issues, both short- and long-term consequences and also all stakeholders' wants and needs into their daily business activities.

The third and last element is the value capture, which may also be addressed as value appropriation [62]. Traditional business model approaches tend to restrict this business element to financial aspects, seeking to evaluate firm's cost structure and revenue model [54]. SBMs include this approach but extends its scope to also include other forms of non-economic value captured. In this sense, a sustainable value capture includes how the company "captures economic value while maintaining or regenerating natural, social and economic capital beyond its organizational boundaries" [40].

*2.2. Assessing Sustainable Value: Corporate Sustainability Performance Dimensions*

One of many PMS frameworks discussed in the literature is the performance prism model, which is composed by five interconnected performance dimensions: stakeholders' satisfaction, strategic drivers, business processes, capabilities and stakeholders' contributions [33]. This approach has the advantage of presenting logical interconnected performance dimensions, being comprehensive and flexible, enabling implicit facts to come to the surface and addressing directly a firm's stakeholders [33]. In the present paper, we apply this framework in the context of corporate sustainability, considering its multi-stakeholder approach. Next, we present a brief discussion of each performance dimension from the perspective of corporate sustainability.

The first dimension has to do with (i) stakeholders' satisfaction. The literature has been using stakeholder theory to justify arguments about corporate sustainability (such as Matos and Silvestre [63] and Perrini and Tencati [64]). By considering stakeholders when assessing corporate sustainability performance, companies are challenged to find solutions in the way that they manage business that are able to promote benefits not only for the firm itself but also for internal and external stakeholders. The literature refers to this combined benefit as shared value [15], win-win solutions [17] or sweet spots [16]. Despite being different concepts with different backgrounds, they all intend to highlight the potential business opportunities that companies have to contribute to sustainable development when making decisions considering not only the economic but also the environmental and social impacts. The second performance dimension refers to (ii) strategic drivers, which have to do with a firm's purpose, mission and corporate values. Therefore, they orient decisions on how to satisfy a firm's internal and external stakeholders. Sustainability strategies can propose new

products and markets [15,65], redefine productivity in the value chain [15,65], build new collaborative value chains [15], etc. Strategic choices for sustainability have direct impact on how companies conduct their (iii) business processes, which is the third performance dimension. Sustainability aspects can be integrated into the management of both primary activities (inbound logistics, production, outbound logistics, marketing and sales, services) and secondary activities (firm infrastructure, human resource management, information and communication technology, procurement) [61,66]. Organizations can implement sustainability initiatives into many business processes, promoting sustainable supply chain management [67,68], eco-design [69,70], sustainable operations management [71], sustainability reporting [46,72], sustainable work design and ergonomics [73], among others. The next performance dimension, (iv) capabilities, considers a combination of people, practices, technology and infrastructure [33]. The literature discusses capabilities that companies need to develop when aiming to be more sustainable. In this sense, knowledge and technology on environmental solutions for products [74] and processes [74,75], as well as skilled human resources motivated by sustainability-oriented leadership [76], are relevant to improve sustainability performance. Moreover, firm's capabilities towards corporate sustainability performance includes several aspects, such as systemic thinking [77,78]; capabilities for learning and developing [76]; capabilities for integrating business, environmental and social issues [76]; change management capabilities [76,79,80]; and networking capabilities [77]. These capabilities need to be dynamic to ensure innovation in the business model for sustainability [81]. The literature suggests that the complex nature of sustainability challenges demands firms to engage different stakeholders towards integrative solutions with the consideration of multi-objectives [63]. Stakeholders, then, have a dual role in corporate sustainability: as targets to understand their needs and desires in order to provide value to them and as means to contribute towards the co-creation of value together with the firm. The last dimension, (v) stakeholders' contribution, can appear in different ways, for instance, bridging business interests with imperatives for community development [82], and promoting partnerships with external stakeholders for research and development cooperation [65,83], such as with suppliers, customers, regulators and communities [26,84,85]. Different group of stakeholder can contribute to the value flow of business models for sustainability [86].

### 2.3. Tool Proposal: Two-Lenses Model for Sustainability Innovations

Combining the SBM elements (discussed in Section 2.1) and the CSP dimensions (discussed in Section 2.2), we propose a Two-Lenses Model (2LM) to support the identification of sustainability innovation opportunities. As represented in Figure 1, the definition of sustainable value proposition is closely related to firm's sustainability performance dimension focused on stakeholders' satisfaction and corporate strategic drivers. This is because these dimensions make explicit its priorities regarding its stakeholders and respective value that the company is seeking to promote. The following three performance dimensions are focused on how the firm intends to create and deliver the proposed sustainable value, integrating capabilities and stakeholders' contributions into firm's business processes. The SBM element of sustainable value capture is represented throughout the five performance dimensions because each performance dimensions promotes value that is expected to be captured when SBMs are put into practice.

By incorporating a process approach into the conceptual framework, the step-by-step process of the 2LM illustrated in Figure 1 can be followed in a structured brainstorming session to identify sustainability innovation opportunities. The proposed sequence to apply 2LM initiates with the definition of the participants and the unit of analysis (Part A). Participants need to have deep knowledge of the company, not only on his or her own area of responsibility but on different business units and strategic and operational aspects. If possible, a multidisciplinary team of internal and external stakeholders would represent the ideal participants set to increase the richness of the discussion. Once the participants are defined, a structured brainstorming session is organized by a moderator. This moderator

needs to have previous knowledge on the 2LM and associated concepts. A large printed version (A2) or projection of the 2LM illustration (Figure 1) is settled so that all participants are able to visualize the figure during the discussions. The brainstorming session initiates with the definition of the business unit to provide clear understanding of the following reflections during the session. This unit can be the organization itself, a specific product line, a network of organizations or any other unit limitation that is in accordance with the participants.

The session follows with Part B (questions 3.1 to 3.5 of Figure 1), with reflections about each performance dimension. This includes the identification of stakeholders and their respective wants and needs (question 3.1), the strategies to promote stakeholders' satisfaction (question 3.2), the business processes to achieve the strategies (question 3.3), the internal capabilities (question 3.4) and stakeholders' contributions to enable these processes. At this stage of the 2LM, participants are asked to reflect about each question and the answers are noted in sticky notes on the corresponding canvas. Part B finishes with question 4, instigating participants in reflections on the relations between the performance dimensions discussed so far. Thus, Part B seeks to make explicit the key aspects of corporate sustainability performance, highlighting the interconnection between the performance dimensions. While describing these aspects, participants tend to start to identify possible strengths and weaknesses of the business model, based on this multidimensional CSP critical analysis.

Part C (questions 5.1 to 5.3) aims to promote reflections on the sustainability innovation opportunities, by rethinking each business model element: sustainable value proposition (question 5.1), value creation and delivery system (question 5.2) and value capture (question 5.3). These reflections retrieve a critical analysis of the performance dimensions associated by the 2LM (Figure 1), as they are registered in sticky notes. Given the expected interrelation between performance dimensions of the prim model [33], one of the discussions during conduction of Part C is about these relations and possible misalignments between dimensions. These may be a source of value destroyed and, by tackling this, business opportunities may be derived [87]. Thus, the combined debate on the SBM elements and the CSP dimensions tend to support unfolding sustainability innovation opportunities for the companies' business models.

To finalize the 2LM application, the structured and moderated brainstorming session ends with a general view of the annotations made in the 2LM illustration and identification of next steps to prioritize efforts for a subsequent action plan to develop further and implement the sustainability innovations opportunities identified.

The tool seeks to provide a view of what is relevant to be measured (from a strategic and systemic view), rather than the measurement of the performance indicators themselves. The tangible measurement indicators can be a step after applying the 2LM, when the action plan based on the sustainable innovation opportunities identified by the tool application is defined and conducted.

## 3. Research Method

The present qualitative research was carried out in two phases. The first phase consisted of a critical analysis of SBM and CSP literature that enabled the proposition of the Two-Lenses Model (2LM) for sustainability innovation, presented in Section 2.3.

The second phase consisted of applying the 2LM in two industrial cases. They were selected according to a theoretical sampling logic (rather than random or stratified) since we used intentional criteria to define the organizations to be studied [88]. Three main selection criteria were used. The first is that the company should be concerned with social and/or environmental goals and not only with financial return. Evidence of this was collected in the companies' websites and during the interviews. The second selection criterion is low level of organizational complexity, e.g., smaller companies were chosen. Since this was the first application of the framework with primary data, we aimed to reduce the difficulty of using the tool by choosing smaller companies with a relatively more controlled

set of variables and interrelations and also an easier access to top management. The third selection criterion is related to the final set of companies, since we aimed for diversified types of companies to enable the exploration of the tool in different scenarios.

Company 1 (C1) is a medium company with a fair-trade signature and counts with suppliers and customers in different countries to commercialize mainly coffee but also cocoa powder and tea. On the other hand, Company 2 (C2) is a capital goods manufacturer that designs and manufactures specialized machines with innovative and customized solutions.

The data were collected from multiple sources of evidence in addition to the interviews [89]. To prepare the interviews, companies' websites were analyzed. For the interviews, the step-by-step process described in Section 2.3 was followed. Thus, a visual representation of the 2LM (Figure 1) and the questions for the interview was brought in an A2 size sheet. The answers were written in sticky notes and directly attached to the A2 alongside with the interview performed directly by the researchers. In this case, the researchers played the moderator role. Two researchers conducted each brainstorming session.

Following the 2LM proposal (Figure 1), the first activity is the definition of participants (Part A). Interviews were performed with the Head of Supply Chain and Procurement in C1 and with the CEO of C2. Notes from discussions with these top managers interviewed were also made during the primary data collection. Parts A, B and C of the 2LM were conducted in both cases, following the questions presented in Figure 1.

Data analysis started with a within-case analysis, providing an in-depth understanding of each company. Key results, including innovation opportunities identified, are presented in Section 4.1. Following this, a cross-case analysis was conducted, as presented in Section 4.2. This enabled the identification of sustainability innovation triggers. This shows evidence that the 2LM tool can serve for the identification of sustainability innovation opportunities, by a critical cross-analysis between SBM elements and CSP dimensions.

## 4. Results and Discussion

### 4.1. Cases' Overview and Identified Innovation Opportunities

This section describes the results achieved in the application of 2LM to identify sustainability innovation opportunities in the industrial cases.

#### 4.1.1. Case 1 (C1)

C1 is an ethical hot drink brand certified by a Fairtrade mark, commercializing mainly coffee but also tea and hot chocolate. This mid-sized company fosters small farmers in different countries, not only buying from them but also by sharing profit with them, involving them in strategic decision making and promoting knowledge exchange and technological advances to be applied to their production activities.

The needs and wants of these stakeholders are taken into consideration by C1: investors, board of shareholders, board of directors, consumers, customers (retailers, export, food service), cooperatives of farmers, farmers, employees, manufacturing partners, foundations, distributors, competitors, environment and society. They also contribute to C1 activities in various ways, such as engagement in social and environmental initiatives, quality service delivery of manufacturing and logistics, provision of human and financial resources. The firm considers three key strategic drivers: focus on the farmers, the integration of environmental action into business and transparency in all their business activities.

Regarding the business processes dimension, C1 points out the following core processes: selection of farmers and procurement, quality management, roasting and packaging, stock management, selling, marketing and community building. The latter represents the emerging distribution channel the firm is building to commercialize high-quality customized coffee. The supporting and complementary processes are the transport of raw material and finished goods, campaigning, research and development, strategic planning and capacity building of farmers. The latter relates to periodic events organized to promote networking between farmers and technical specialists in the field of coffee, tea and cocoa,

aiming at improving the growing process of the farmers. Regarding C1's capabilities, the capacity to identify small farmers to become new partners was mentioned, together with building and fostering the relationship with them. This allows the firm to have access to a broad range of farms, with a direct and transparent bond. Another important capability developed by C1 is the marketing and selling skill to manage the relationship with customers and consumers. This is directly linked to one of the most important assets of the firm, which is its brand.

One identified opportunity from applying 2LM is related to the misalignment between the capability and business process performance dimensions. On the one hand, the firm has great capability in its fair-trade procurement process, including knowledge on and processes of managing transparency in the supply chain and a broad network of small farmers. This is currently used by the company to supply its own products. However, during the interview using the 2LM, it was mentioned that more value could be created from this capability, including new business opportunities, e.g., by providing services to support other companies on how to manage supply chains according to fair-trade guidelines. This service could be offered both to providers of other non-coffee products and to competitors alike. As consequence, a potential secondary positive effect would be an increased stability in production volume for farmers, thus positively impacting their operations planning.

Other business opportunities were derived from alternatives to increase stakeholders' satisfaction. One example of this is the potential expansion into the food sector to provide their B2B customers, e.g., coffee shops, with more integrated solutions. This may be achieved by broadening their product mix with the inclusion of offerings such as food, coffee machinery, stock management service of consumable goods, etc. These demands broaden C1's capabilities and business processes with direct impact on the firm's system of value creation and delivery, since new offerings would be part of the value proposition.

### 4.1.2. Case 2 (C2)

C2 is small-sized capital goods provider, family owned, which designs and manufactures equipment with high innovative content. C2 is not exclusively driven by financial results. On top of that, its owner seeks to create social value, such as the satisfaction of working in challenging tasks towards innovative solutions, the opportunity to provide income for another employee and the satisfaction of continuing the family's legacy. Regarding the stakeholders' satisfaction dimensions, it is interesting to note the importance to satisfy the employees' needs, given the firm's dependency on them. This is ensured not only by financial recognition (salary) but also by the satisfaction to overcome technical challenges of the clients, to constantly push towards creativity and innovation and to be able to conduct from idea to physical solution, amongst others. According to the interviewee, the type of machinery produced has very low energy demand and produces low levels of waste. Therefore, environmental aspects are not considered critical in the firm's operations.

The firm does not have formally stated strategic drivers. However, strategic priorities were pointed out by the owner: ensure high quality and innovation for products and services, growth to increase to number of employees and cultivate partnerships with clients and suppliers. These three aspects were mentioned many times during the interview, reinforcing their importance to C2. About the business processes dimension, it includes research of engineering solutions, product design, manufacturing, procurement, customer relations, after sales services and finance, amongst others. The most relevant capabilities for the firm are creativity, technical capacity and market knowledge. They are fundamental to guaranteeing C2's competitive advantage and client satisfaction. The last performance dimension is related to the stakeholders' contributions. These are very aligned with the aspects highlighted so far, with special focus on employees and partnerships with clients and suppliers.

One misalignment identified is regarding firm's technical knowledge that is not yet explored towards eco-efficiency services. By addressing this misalignment between firm's

capability and business process, C2 may increase its value proposition with additional services, by focusing on increasing value creation and delivery from this knowledge. On the stakeholders' satisfaction innovation trigger, C2 has the opportunity to promote network building with other micro and family-owned companies, given their very particular set of conditions and situations. Possible informal meetings may contribute to more value creation from experience sharing, since lessons learned in one organization could be used in another one.

### 4.2. Unfolding Sustainability Innovation Opportunities

Based on the data collected, the analysis of performance dimensions enabled the identification of innovation opportunities to improve sustainable value exchange with stakeholders. These opportunities are a combination between novelty in terms of relationship with stakeholders and of access to the necessary capabilities for innovation. The first aspect represents whether the stakeholder is currently being addressed by the firm or not and whether the relation with the firm remains the same or will be further developed. The second aspect indicates whether the sustainability opportunity builds on current capabilities or if new ones need to be developed. Table 2 shows examples of innovation opportunities from the case studies considering these aspects. It also indicates the primary element of the business model that would change by implementing the innovation.

**Table 2.** Innovation opportunities to increase sustainable value exchange with stakeholders.

| | Current Relations with Stakeholders | New Relations with Current Stakeholders | New Stakeholders |
|---|---|---|---|
| Existing Capabilities | Deepener relationship with customers and strengthen brand with Coffee Club members (VC, C1) | Commercialize procurement service of fair-trade products and of design of transparent supply chains (VP, C1) Add services on eco-efficiency solutions for equipment (VP, C2) Foster partnership with suppliers for risk sharing (deadline to client) (VD, C2) | Network building for knowledge exchange (VC, C2) |
| New Capabilities | Reduce environmental impact of packaging with more research and development (VC, C1) | Develop a broader offering for the food service customer segment (VP, C1) | Open a coffee shop branded by C1 (VP, C1) |

Note: VP = Value proposition; VD = Value creation & delivery system; VC = Value Capture; C1 = Case 1; C2 = Case 2.

During the conduction of the 2LM, two internal triggers to identify the sustainability innovation opportunities were identified: the misalignment between sustainability performance dimensions and the gaps in stakeholders' satisfaction. The first trigger is based on verifying whether a particular performance dimension is aligned with the others. This includes possible guiding questions such as: (a) Are business process and capabilities performed in a way that contributes to deliver the business strategy? (b) Are stakeholders' contributions enough to enable business processes? (c) Do firm's capabilities suffice to put the firm's strategic drivers into practice?

Misalignment inefficiencies between actors in the supply chain were previously pointed out by the literature as potential issues to unlock business model innovation [87]. Our research complements these authors' research by showing empirical evidence that internal misalignments (rather than inter-organizational ones) can also provide insights for business model innovation. For instance, exploring new ways to convert a firm's capability into value can be a source of competitive advantage [90,91]. Aligned to this, we found empirical evidence that business innovation opportunities can be identified from internal misalignments between capabilities and business process performance dimensions. For example, C1 has the potential to explore its capabilities on transparent supply chain management by implementing new business processes to provide services in this area. In turn, C2's technical capabilities and innovation skills can be further used for promoting more eco-efficient services.

The second internal trigger for more sustainable business models is focused on the stakeholders' satisfaction performance dimensions, seeking to reveal stakeholders' wants and needs that are not being satisfied by the company and reflecting on the reason for dissatisfaction. The literature suggests the involvement and engagement of users to derive opportunities to generate new ideas for sustainable business innovations [92]. In addition, different approaches may be defined to address the various corporate stakeholders, including reinforcement strategies for supportive stakeholders, stabilization strategy for passive stakeholders (when passive communication is considered sufficient) or containment strategies for stakeholders with potential to obstruct business operations [93]. To foster interaction with users, C1 identified the opportunity to intensify relations with members of their exclusive club, as well as to enlarge solutions for service customer segment. By doing this, the company may gather knowledge and experience to open its own coffee shop, thus, getting ready for a value chain vertical integration. It is worth noting that corporate sustainability is about satisfying not only customers and shareholders but also other stakeholders (such as employees, communities, pressure groups, etc.) [11]. In this sense, for C2, it is fundamental to promote employees' satisfaction [7], which may be fostered by building a network with other family-owned companies to share experience and lessons learned. As pointed out previously in the literature, there are business opportunities in exploring social and ecological problems associated with stakeholders' of the organization [94]. A sustainable business is expected to have a positive effect not only to a specific niche but rather to the market in general and to society [95]. These stakeholders also need to be involved in order to achieve true sustainable business models [96].

Both literature analysis and case studies converge in the dynamics of the two lenses of CSP and SBM. Figure 2 illustrates this. The present research found insights that the critical analysis of the company's CSP dimensions can be used as a trigger for innovation opportunities for more sustainable business models. These opportunities include exploring new offerings to improve sustainable value proposition; identifying new mechanisms for more sustainable value creation and delivery systems; and fostering positive impacts with more efficient and effective sustainable value capture. Independently from which trigger the opportunity is derived, the research also shows that the business innovation opportunities can be primarily closer related to one element of the business model (value propositions, value creation and delivery system or value capture). In turn, this innovation implicates also changes in the other elements since they are directly interconnected. In addition, in a focused brainstorming section, the 2LM also reinforces the importance of retrieving companies identity and creativity, which have been shown to be directly related to green innovations [97].

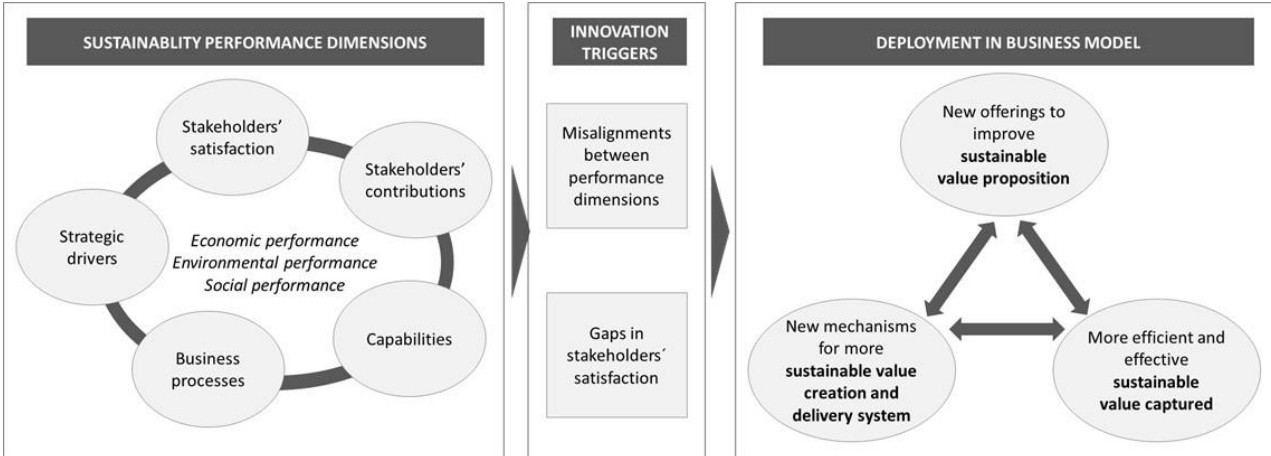

**Figure 2.** Sustainability innovation triggers and impact on business model elements.

## 5. Conclusions

This research provides three main contributions. First, we bridge the CSP and SBM lenses, as they have in common the fact that both can be used to frame an organization, supporting it in its decision-making process. The main difference between the lenses lies in what they emphasize. While the SBM lens focuses on showing the mechanisms (using sustainability innovations) to intensify sustainable value proposition, creation, delivery and capture, the CSP lens is adequate to evaluate the gap between the current and aimed situation of the company's actions and results. Thus, CSP is more closely related to evaluation criteria, indicators and parameters. We then argue that both lenses are complementary and contribute to each other. Second, our research shows that the two conceptual models of CSP and SBM can be operationalized from an integrative perspective. This research argues this by proposing a tool called 2LM for sustainability innovations. It seeks to support companies in reflecting on their performance dimensions from a sustainability perspective. In particular, company participants confirmed that the 2LM helped them to find interesting implications, since it provides a structured way to rethink their business models towards sustainability innovations. The analysis of tool use and findings from the case studies led to the third research contribution, which is the identification of two triggers for sustainability innovations opportunities: (1) misalignment between performance dimensions and (2) gaps in stakeholder satisfaction. These triggers can be further explored by academics and managers that seek to promote sustainability innovation opportunities.

Despite its contributions, the research also presents some limitations. Regarding the 2LM, there is a trade-off between the level of detail and the ease of comprehending a whole overview of the information in only one figure. The use of the model in the cases also showed that sustainable development challenges were not explicitly brought by the process. Additionally, the model tends to focus on sources of ideas for sustainability innovation that are based on internal issues (company's performance dimensions) and/or on the internal perception of stakeholders' wants and needs.

These limitations on the 2LM call to be addressed for future studies, evolving the tool to explicitly include the sustainable development challenges and the active participation of external stakeholders. There is a need to expand the process management perspective relating to the increasing pressure from customers and their requirements introduced in the system [98]. Additionally, there is an opportunity for the further investigation and deployment of tangible measurement indicators linked with the action plan based on the sustainable innovation opportunities identified by the 2LM. Finally, the model is also limited due to absence of aspects related to organizational culture and values, which are particularly relevant for strategic decisions in the corporate sustainability context [99].

**Author Contributions:** Conceptualization, S.N.M. and M.H.; Formal analysis, S.N.M. and M.H.; Investigation, S.N.M. and M.H.; Methodology, S.E. and M.M.C.; Project administration, S.N.M.; Supervision, S.E., M.M.C. and I.B.; Validation, S.E.; Visualization, M.M.C. and P.R.J.; Writing—original draft, S.N.M. and M.H.; Writing—review & editing, P.R.J. and I.B. All authors have read and agreed to the published version of the manuscript.

**Funding:** This research was funded by CAPES (Brazilian Coordination of Superior Level Staff Improvement) and CNPq (Brazilian National Council for Scientific and Technological Development).

**Institutional Review Board Statement:** Not applicable.

**Informed Consent Statement:** Not applicable.

**Acknowledgments:** The authors would like to thank the Brazilian National Council for Scientific and Technological Development—CNPq, Brasilia—Brazil; the Paraiba Research Funding Foundation—FAPESQ, Campina Grande—Brazil; and the Coordination for the Improvement of Higher Education Personnel—CAPES, Brasilia—Brazil.

**Conflicts of Interest:** The authors declare that there are no conflicts of interest regarding the publication of this paper.

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
