# Peer review of "Two-Lenses Model to Unfold Sustainability Innovations: A Tool Proposal from Sustainable Business Model and Performance Constructs"

_sustainability, doi:10.3390/su14010556_

Round 1
Reviewer 1 Report
The research on “Two-Lenses Model to Unfold Sustainability Innovations: A Tool Proposal from Sustainable Business Model and Performance Constructs” aims to investigate how corporate sustainability performance (CSP) and sustainable business models (SBMs) concepts can be combined to support companies in identifying innovations for sustainability and propose a step-by-step process-based tool called Two-Lenses Model (2LM).
1). Your problem statement in the Abstract “Previous research identified corporate sustainability performance and sustainable business models as separate perspective of analysis towards encouraging sustainability innovation and sustainable value creation to multiple stakeholders.” seems contrasting with the following Article:
+ The sustainability performances of sustainable business models: https://doi.org/10.1016/j.jclepro.2021.129145
I want to hear your explanation/clarification on this matter.
2). Also, I suggest adding sentence(s) to link the two following sentences in the Abstract:
This research investigates how corporate sustainability performance (CSP) and sustainable business models (SBMs) concepts can be combined to support companies to identify innovations for sustainability.
….
We propose a step-by-step process-based tool called Two-Lenses Model (2LM)….
The connecting sentence(s) should present the state of art towards proposing a step-by-step process-based tool called Two-Lenses Model (2LM).
3). In general, the research methods and results are not clearly presented in the Abstract.
4). At the beginning of the Introduction section, you jumped to present on the aims at maximizing profit while promoting social and environmental values of the companies these days, which is a compelling idea and many times hard to put into practice and to figure out multiple approaches on sustainability and sustainable development exist without a common ground. As your research focused on sustainability/sustainable development aspects, it would be better to present briefly sustainable development background linked to the current trends on sustainable development goals (SDGs) at the beginning before getting focused on your topics of CSP and SBMs. I suggest adding a paragraph to present this brief background by indicating each of economic (profit), social, and environmental values. I recommend the two following articles for your references:
+ The sustainability performances of sustainable business models: https://doi.org/10.1016/j.jclepro.2021.129145
+ Assessing Sustainability of the Capital and Emerging Secondary Cities of Cambodia... https://doi.org/10.3390/data5030079
5). I suggest expanding the explanation in 155-157 to the importance or the reasons of proposing a step-by-step process to support the identification of opportunities for sustainability innovations before moving to each division (its sub-sections).
6). I suggest presenting Figure 1 in the Research Methods section and more explaining this figure by connecting it to the explanation of your research methods.
7). Based on the nature of your current Results and Discussion sections, I suggest combining these two sections (5. Results and Discussion). Its subtitles should be put as sub-sections:
5.1. Case Overview and Identified Innovation Opportunities
….....
5.2. Unfolding Sustainability Innovation Opportunities
8). After all, I suggest revising your conclusions to be short and clear. The conclusions in a scientific article should describe the usefulness of the results in the field of research and of course, be limited to the specific area of research investigated. Therefore, no need to re-explain your research objectives/questions and methods in the conclusions.
Author Response
Dear Reviewe,
We do appreciate the time and effort you put in revising our manuscript. We improved the article that significantly contributed to improving our manuscript.
The modification are highlighted in yellow in the manuscript. Bellow you can find our modifications regarding each comment you presented.
We sincerely hope you will appreciate reading the revised manuscript and consider our improvements sufficient for publication in the Sustainability.
Best regards,
Prof. Dr. Ivan Bolis on the behalf of the authors
The research on “Two-Lenses Model to Unfold Sustainability Innovations: A Tool Proposal from Sustainable Business Model and Performance Constructs” aims to investigate how corporate sustainability performance (CSP) and sustainable business models (SBMs) concepts can be combined to support companies in identifying innovations for sustainability and propose a step-by-step process-based tool called Two-Lenses Model (2LM).
|
1). Your problem statement in the Abstract “Previous research identified corporate sustainability performance and sustainable business models as separate perspective of analysis towards encouraging sustainability innovation and sustainable value creation to multiple stakeholders.” seems contrasting with the following Article: + The sustainability performances of sustainable business models: https://doi.org/10.1016/j.jclepro.2021.129145 I want to hear your explanation/clarification on this matter. |
We started developing this article in 2020 and this article is not yet published. We removed this sentence from the abstract.
Regarding the reference, we consider it in the introduction, introducing the following sentence: “A recent study [20] discussed how SBMs are effectively driving sustainable performance, identifying some tensions and conflicts between the three pillars of sustainability when developing and implementing SBMs. This points to the need to investigate this connection.”
|
|
2). Also, I suggest adding sentence(s) to link the two following sentences in the Abstract: This research investigates how corporate sustainability performance (CSP) and sustainable business models (SBMs) concepts can be combined to support companies to identify innovations for sustainability. …. We propose a step-by-step process-based tool called Two-Lenses Model (2LM)…. The connecting sentence(s) should present the state of art towards proposing a step-by-step process-based tool called Two-Lenses Model (2LM).
|
We improved the abstract and we introduced more details considering some modifications of the paper |
|
3). In general, the research methods and results are not clearly presented in the Abstract.
|
We improved the abstract considering your suggestions. |
|
4). At the beginning of the Introduction section, you jumped to present on the aims at maximizing profit while promoting social and environmental values of the companies these days, which is a compelling idea and many times hard to put into practice and to figure out multiple approaches on sustainability and sustainable development exist without a common ground. As your research focused on sustainability/sustainable development aspects, it would be better to present briefly sustainable development background linked to the current trends on sustainable development goals (SDGs) at the beginning before getting focused on your topics of CSP and SBMs. I suggest adding a paragraph to present this brief background by indicating each of economic (profit), social, and environmental values. I recommend the two following articles for your references: + The sustainability performances of sustainable business models: https://doi.org/10.1016/j.jclepro.2021.129145 + Assessing Sustainability of the Capital and Emerging Secondary Cities of Cambodia... https://doi.org/10.3390/data5030079
|
We improved the first part of the introduction section (in yellow) with your suggestions. Besides, we introduced and discussed the two suggest references. |
|
5). I suggest expanding the explanation in 155-157 to the importance or the reasons of proposing a step-by-step process to support the identification of opportunities for sustainability innovations before moving to each division (its sub-sections).
|
As suggested, we introduced a paragraph above Table 1, improving the explanation of the step-by-step process. |
|
6). I suggest presenting Figure 1 in the Research Methods section and more explaining this figure by connecting it to the explanation of your research methods.
|
Figure 1 is the result of the literature review presented in section 2. We were unable to change its position for the methodology chapter, but we have greatly improved the text to reduce the need to have this figure in this section 3.
|
|
7). Based on the nature of your current Results and Discussion sections, I suggest combining these two sections (5. Results and Discussion). Its subtitles should be put as sub-sections: 5.1. Case Overview and Identified Innovation Opportunities …..... 5.2. Unfolding Sustainability Innovation Opportunities
|
We accepted and implemented your suggestion |
|
8). After all, I suggest revising your conclusions to be short and clear. The conclusions in a scientific article should describe the usefulness of the results in the field of research and of course, be limited to the specific area of research investigated. Therefore, no need to re-explain your research objectives/questions and methods in the conclusions.
|
We revised the conclusion section to include only the main contributions and limitations of the article. It is now concise and hopefully clearer. |

Reviewer 2 Report
It is a great honor to have the opportunity to review this manuscript. Sustainable innovation is crucial for social development. Based on this, the author uses case analysis to make a beneficial exploration. In general, the research has some new ideas and logical structure is reasonable. It is suggested to publish after minor revision, with specific comments as follows:
(1) The marginal contribution of research is not clear. Though the authors propose a step-by-step process-based tool called Two-Lenses Model (2LM), it is innovative to suggest that more words be used to describe why it is important to do so.
(2) The analysis section recommends more detail. For example, part 3.3 is one of the core contents of this study, with relatively few analyses and more detailed suggestions.
(3) Two Industrial Cases selection standard recommendations are introduced in more detail. At the same time, it is suggested to further expand the discussion, and further introduce the generalizable experience and limitations of these two cases.
Author Response
Dear Reviewer,
We do appreciate the time and effort you put in revising our manuscript. We improved the article that significantly contributed to improving our manuscript.
The modification are highlighted in yellow in the manuscript. Bellow you can find our modifications regarding each comment you presented.
We sincerely hope you will appreciate reading the revised manuscript and consider our improvements sufficient for publication in the Sustainability.
Best regards,
Prof. Dr. Ivan Bolis on the behalf of the authors
It is a great honor to have the opportunity to review this manuscript. Sustainable innovation is crucial for social development. Based on this, the author uses case analysis to make a beneficial exploration. In general, the research has some new ideas and logical structure is reasonable. It is suggested to publish after minor revision, with specific comments as follows:
|
(1) The marginal contribution of research is not clear. Though the authors propose a step-by-step process-based tool called Two-Lenses Model (2LM), it is innovative to suggest that more words be used to describe why it is important to do so. |
The tool's contributions have been included in the first part of the new version of the conclusion section. |
|
(2) The analysis section recommends more detail. For example, part 3.3 is one of the core contents of this study, with relatively few analyses and more detailed suggestions. |
We consider these suggestions. In particular we expanded the contents of chapter 3.3 (current chapter 2.3) |
|
(3) Two Industrial Cases selection standard recommendations are introduced in more detail. At the same time, it is suggested to further expand the discussion, and further introduce the generalizable experience and limitations of these two cases. |
The relevance of this article is the proposal of the Two-Lenses Model (2LM). Both cases allowed us to apply this tool. Each case has context-sensitive data, but in section "4.2 Unfolding sustainability innovation opportunities," we discussed some generalizations. In particular, through cross-case analysis, we show two internal triggers to identify the sustainability innovation opportunities: the misalignment between sustainability performance dimensions and the gaps in stakeholders' satisfaction. We also introduced the limitations of these two cases in the second part of the conclusion section
|

Reviewer 3 Report
I recommend making the abstract more specific so that it reflects the particular contributions of the article.
The literature review is well-conceived and mentions several prominent authors. However, I would recommend including insights into a business through processes, strategy, performance and also include relevant publications in sustainability:
1. Rolinek, L. et al.: Level of Process Management Implementation in SMEs and Some Related Implications. Transformations in Business and Economics 2015, 14, 360-377.
doi.org/10.3390/su8111212
doi.org/10.1016/S2212-5671(14)00394-3
doi.org/10.3390/su8010043
A challenging part of the article is the vague definition of its objective, which is not clearly defined in the text.
A significant negative of the article which relies on performance in many cases is the unmeasurability of the results of the two enterprises. I find this to be a major issue in terms of models for management. There is a need to expand more on this area to make the suggestions less general and more graspable.
Author Response
Dear Reviewer,
We do appreciate the time and effort you put in revising our manuscript. We improved the article that significantly contributed to improving our manuscript.
The modification are highlighted in yellow in the manuscript. Bellow you can find our modifications regarding each comment you presented.
We sincerely hope you will appreciate reading the revised manuscript and consider our improvements sufficient for publication in the Sustainability.
Best regards,
Prof. Dr. Ivan Bolis on the behalf of the authors
Comments and Suggestions for Authors
|
I recommend making the abstract more specific so that it reflects the particular contributions of the article. |
We improved the abstract (in yellow) |
|
The literature review is well-conceived and mentions several prominent authors. However, I would recommend including insights into a business through processes, strategy, performance and also include relevant publications in sustainability:
|
We improved the introduction section, and we included this publication
|
|
A challenging part of the article is the vague definition of its objective, which is not clearly defined in the text |
We improved this part in the introduction section with this text: “The main objective of this article is to propose and apply a tool to combine CSP and SBM to support identifying sustainable innovation opportunities. Providing a more systemic business perspective, this tool seeks to enable a structured brainstorming based on CSP and SBMs concepts to support the identification of sustainability innovation opportunities that include a strong multi-stakeholder perspective.” We also improved many other parts of the article to make this objective clearer.
|
|
A significant negative of the article which relies on performance in many cases is the unmeasurability of the results of the two enterprises. I find this to be a major issue in terms of models for management. There is a need to expand more on this area to make the suggestions less general and more graspable. |
The tool seeks to provide a view of what is relevant to measure from a strategic and systemic perspective rather than measuring the performance indicators themselves. The research does not present measurable results of the two enterprises, as this was not the research objective and was not contemplated in the research design. The tangible measurement indicators can be a step after applying the 2LM, when the action plan based on the sustainable innovation opportunities identified by the tool application is defined and conducted. Following your suggestion, we included this argument at the end of Section 2 to provide a more explicit alignment of expectations, and we return to this point in future research agenda. |

Round 2
Reviewer 1 Report
The authors have addressed my concerned comments.
Reviewer 3 Report
I recomend this article, authors impreved the paper